# *Aggregatibacter Actinomycetemcomitans*: Clinical Significance of a Pathobiont Subjected to Ample Changes in Classification and Nomenclature

**DOI:** 10.3390/pathogens8040243

**Published:** 2019-11-18

**Authors:** Niels Nørskov-Lauritsen, Rolf Claesson, Anne Birkeholm Jensen, Carola Höglund Åberg, Dorte Haubek

**Affiliations:** 1Department of Clinical Microbiology, Aarhus University Hospital, DK-8200 Aarhus N, Denmark; nielnoer@rm.dk; 2Department of Odontology, Division of Oral Microbiology, Umeå University, S-901 87 Umeå, Sweden; rolf.claesson@umu.se; 3Department of Dentistry and Oral Health, Aarhus University, DK-8000 Aarhus C, Denmark; abj@dent.au.dk; 4Department of Odontology, Division of Molecular Periodontology, Umeå University, S-901 87 Umeå, Sweden

**Keywords:** adherence, endocarditis, fimbriae, JP2, leukotoxin, periodontitis

## Abstract

*Aggregatibacter actinomycetemcomitans* is a Gram-negative bacterium that is part of the oral microbiota. The aggregative nature of this pathogen or pathobiont is crucial to its involvement in human disease. It has been cultured from non-oral infections for more than a century, while its portrayal as an aetiological agent in periodontitis has emerged more recently. *A. actinomycetemcomitans* is one species among a plethora of microorganisms that constitute the oral microbiota. Although *A. actinomycetemcomitans* encodes several putative toxins, the complex interplay with other partners of the oral microbiota and the suppression of host response may be central for inflammation and infection in the oral cavity. The aim of this review is to provide a comprehensive update on the clinical significance, classification, and characterisation of *A. actinomycetemcomitans*, which has exclusive or predominant host specificity for humans.

## 1. Introduction

*Aggregatibacter actinomycetemcomitans* is the type species of genus *Aggregatibacter*, which is part of bacterial family *Pasteurellaceae*. [*Bacterium actinomycetem comitans*] was cultured from actinomycotic lesions of humans in the early 20th century. The absence of related microorganisms rendered it difficult to classify this Gram-negative, fastidious rod, and isolates cultured from invasive infections were referred to national reference institutions. The expanding field of oral microbiology with a focus on periodontitis, particularly the localized, severe form that affects adolescents, caused a renewed interest in the bacterial species. In 2006, the current species name was adopted, and *A. actionomycetemcomtians* became type species of a new bacterial genus, *Aggregatibacter*. Influential events in the narrative of *A. actinomycetemcomitans* are listed in Table 1.

*A. actinomycetemcomitans* is one species among a plethora of microorganisms that constitute the oral microbiota. It has been estimated that at least 500 different bacterial species colonise the oral cavity [13,14,15], and half of these may have been cultivated and validly named because of vigorous efforts directed to the cultivation of oral bacteria. Analysis of a large number of 16S rRNA gene clones from studies of the oral microbiota increased the number of taxa to 619 [16], and the number is steadily increasing (www.homd.org). Bacterial species cannot be validly named in the absence of a cultured type strain [17]. Although “taxa”, “phylotypes” or “operative taxonomic units” revealed by deep sequencing of polymerase chain reaction (PCR)-amplified 16S rRNA genes have relevance for recognition of microbial fluctuations in health and disease, only cultivable microbiota can be made subject to extensive characterisation, including adherence, animal experiments, antimicrobial susceptibility, co-culture, generation of mutants, and growth characteristics.

Carriage of *A. actinomycetemcomitans* appears to be highly host-specific. Although the spread and dissemination of bacterial clones occur, these are not frequent events; hosts tend to carry their strain from teething to edentulous old age [18]. Yet, the species encompasses properties that sometimes reveal its significance in human disease. Particularly, a single serotype b clonal lineage designated the JP2 clone is associated with a severe form of localised periodontitis and tooth loss in adolescents [12]. But rather than being the causative agent of aggressive periodontitis, *A. actinomycetemcomitans* may be necessary for the action of a consortium of bacterial partners by suppressing host defences [19]. It may be classified as a low abundance oral pathobiont, defined as a member of the microbiota that exerts specific effects on the host’s mucosal immune system associated with the development of disease [20]. Although *A. actinomycetemcomitans* may accompany (*comitans*) *Actinomyces*, the narrative of a pathobiont is not valid for other invasive infections such as infectious endocarditis, where *A. actinomycetemcomitans*—when identified—is detected as the sole pathogen by culture and/or PCR on removed heart valves. Severe periodontitis and infective endocarditis are two prominent diseases of very different prevalence, symptoms, and outcome. Although they may share a causative microorganism, a number of conditions is still unknown, and host factors, oral hygiene, and incidental circumstances may be instrumental.

The aim of the present review is to provide a comprehensive update on the characterisation, classification and clinical significance of *A. actinomycetemcomitans* with a particular focus on selected clinical entities. Adhesion, persistence, and inactivation of immune cells are probably essential for the understanding of the intimate association with the host, and these factors are detailed for the purpose of the elucidation of pathogenicity. A number of relevant publications and reviews of other important biochemical mechanisms of this bacterial species are listed in the relevant sections.

## 2. Taxonomy, Classification, Serotype (St) and Population Structure

More than 100 years ago, [*Bacterium actinomycetem comitans*] was co-isolated with *Actinomyces* from actinomycotic lesions of humans [1] (*Actinomyces*, ray fungus, referring to the radial arrangement of filaments in *Actinomyces bovis* sulfur granules; actinomycosis, a chronic disease characterised by hard granulomatous masses). Ample changes have occurred in the classification and nomenclature of this species. Despite the limited similarity with *Actinobacillus lignieresii*, it was reclassified as [*Actinobacillus actinomycetemcomitans*] in a seminal textbook from 1929 [2]. According to Cowan [21], the bacterium was placed in this genus because ‘neither Topley nor Wilson could think where to put it’. In 1962 the phenotypic resemblance of [*Actinobacillus actinomycetemcomitans*] with [*Haemophilus aphrophilus*] was described [3], and a subsequent relocation of [*Actinobacillus actinomycetemcomitans*] to genus *Haemophilus* occurred [22]. Nomenclatural classification as [*Haemophilus actinomycetemcomitans*] within the genus *Haemophilus* permitted antimicrobial susceptibility testing according to standards outlined by the US Clinical and Laboratory Standards Institute. Disk diffusion could be performed and interpreted on *Haemophilus* test medium (HTM) in 5% CO_2_, and HTM broth microdilution testing was carried out in ambient air [23]. However, the nomenclatural relocation did not result in a satisfying classification, because neither [*Actinobacillus actinomycetemcomitans*] nor [*Haemophilus aphrophilus*] are adequately related to *Haemophilus influenzae*, the type species of the genus *Haemophilus*. Finally, in 2006 the new genus *Aggregatibacter* was created to accomodate *Aggregatibacter actinomycetemcomitans*, *Aggregatibacter aphrophilus* and *Aggregatibacter segnis* [11]. A fourth *Aggregatibacter* species, *Aggregatibacter kilianii*, has recently been named (Figure 1) [24].

In the early 1980s, three distinct surface antigens of *A. actinomycetemcomitans* were identified [9], while six serotypes (a through f) were recognised by 2001. The serological specificity is defined by structurally and antigenically distinct O-polysaccharide components of their respective lipopolysaccharide molecules. A seventh St, designated St(g), with a 1:1 ratio of 2,4-di-O-methyl-rhamnose and 2,3,6-tri-O-methyl-glucose, was recently added [25]. St(a), St(b), and St(c) are globally dominant [26], but the distribution may vary according to ethnicity and geography. In Scandinavia, the three dominant serotypes are equally prevalent, while predominance of St(c) is observed in Chinese, Japanese, Korean, Thai and Vietnamese populations [27,28,29,30,31]; a noticeable high prevalence of St(e) has been reported among Japanese periodontitis patients [32]. Assessment of serotype-specific antibodies supports these findings, as all early-onset periodontitis patients from Turkey and Brazil had elevated antibody levels to St(c) and St(a), while St(b) levels were higher in the US [33,34]. 

An initial characterisation of the population structure of the species was published in 1994, using multi-locus enzyme electrophoresis [35]. Two large and four small divisions were identified, with division I (St(a) and St(d)) and III (St(b) and St(c)) encompassing 34% and 58% of the 97 strains analysed, respectively. Two St(e) strains occupied separate divisions (II and VI), one St(c) strain constituted electrophoretic division IV, while division V was composed of two St(a) and one non-serotypable strain. Sequencing of a 16S rRNA gene fragment from 35 strains suggested a different structure with three major clusters [36]. RNA cluster I included 12 strains of four serotypes (a, d, e, and f), all 10 St(b) strains belonged to RNA cluster II, while RNA cluster III only included St(c) strains (N = 10). Strains of particular serotypes were not exclusively confined to specific RNA clusters; one St(a) strain belonged to the St(b) cluster (II), and two divergent RNA clusters were composed of single strains, namely a St(c) (RNA cluster IV) and a St(e) strain (RNA cluster V), respectively [36]. 

One study attempted to establish a multi-locus sequence typing (MLST) scheme for *A. actinomycetemcomitans* [37]. Six gene fragments from the *Haemophilus influenzae* MLST scheme were used. The investigation focused on the JP2 clone, which contributed 66 of 82 strains. MLST has insufficient power to reveal dissemination patterns of clonally related strains, and point mutations of two pseudogenes present in the JP2 clone were more versatile in this respect [37]. Moreover, a MLST web site was not organised and, therefore, the benefits of a portable typing scheme were not corroborated. But MLST of 16 non-JP2 strains carefully selected from the enzyme electrophoresis study [35] suggested the existence of four phylogenetic clusters, rooted by an outgroup consisting of an uncommon St(e) strain. Two related clusters were composed of St(b) and St(c) strains, respectively, while a more distinct cluster encompassed strains of St(a), St(d) and St(e) [37].

Restriction fragment length polymorphism using various restriction enzymes and arbitrarily-primed PCR has been used to differentiate types of *A. actinomycetemcomitans* cultured from patients with severe periodontitis and healthy controls [30,37,38,39,40,41,42]. The method is versatile and discriminative, but lacks portability and a common nomenclature; thus, it is of value for individual studies of specific strains, but lacks general applicability and descriptive significance.

Finally, whole genome sequencing has been introduced for characterisation of the species [43,44,45]. In the largest study, sequences from two human strains of *Aggregatibacter aphrophilus*, 30 human *A. actinomycetemcomitans* strains, and one St(b) strain isolated from a rhesus macaque Old World monkey were used for selection of 397 core genes which were concatenated and trimmed to produce a single alignment of 335,400 bp [45]. Five clades were recognised, designated clade b, clade c, clade e/f, clade a/d and clade e’. Although the analysis clearly separated six strains of serotype b from six strains of serotype c, a close similarity was observed between these two clades, as well as between clade a/d and e/f. In contrast, the clade designated e’, encompassing four St(e) strains, was phylogenetically distinct. The open reading frames necessary for expression of St(e) antigen were highly conserved between clade e and clade e’ strains, but e’ strains were found to be missing the genomic island that carries genes encoding the cytolethal distending toxin. Moreover, the clade e’ strains were more related to an Old World primate strain and carried the unusual 16S rRNA type V sequence (RNA types as defined by Kaplan et al. [36]). Although bacterial species are not defined by DNA sequence, average nucleotide identity (ANI) values locate whole genome sequences from this group/clade outside the species boundary [44]. Thus, strains belonging to the so-called clade e’ (as well as the rhesus macaque monkey strain) may possibly be transferred to new species, and *A. actinomycetemcomitans* may be restricted to strains with exclusive host specificity for humans.

A recent study compared whole genome sequences of strains from blood stream infections supplemented with oral reference strains [46]. Exclusion of so-called clade e’ strains increased the number of core genes present in all strains from 1146 to 1357. Strains of *A. actinomycetemcomitans* are basically divided into three lineages (numbering of lineages differs from reference [44]). Lineage I encompasses the type strain and consists of two groups (St(b) and St(c), respectively). Lineage II consists of St(a) plus St(d)-(g). In contrast to lineage I, many strains of different serotypes from this lineage are competent for natural transformation, and the average size of genomes is approximately 10% larger than in lineage I. Lineage III also expresses St(a) membrane O polysaccharide, and the genome size is comparable to lineage II. However, all six investigated strains were incompetent for transformation due to inactivation of multiple competence genes [46]. 

In conclusion, St designations are valuable for initial typing of clinical strains, but insufficient for characterisation. Recognition of a general MLST scheme could be helpful, and whole genome sequences could be used for MLST and in silico serotyping, as well as further characterisation and epidemiologic investigations. The species description consisting of three separate lineages is figurative, but more knowledge on the new lineage III is needed to disclose the relevance for phenotype, host specificity and pathogenicity.

## 3. General Characteristics 

*A. actinomycetemcomitans* is a fastidious, facultatively anaerobic, non-motile, small Gram-negative rod, 0.4–0.5 µm × 1.0–1.5 µm in size. Microscopically, the cells may appear as cocci in broth and in clinical samples. It grows poorly in ambient air, but well in 5% CO_2_ [47]. Colonies on chocolate agar are small, with a diameter of ≤0.5 mm after 24 h, but may exceed 1–2 mm after 48 h [48]. Primary colonies are rough-textured and adhere strongly to the surface of agar plates (Figure 2).

### 3.1. Recovery, Phenotype, and Molecular Detection

Relevant sites in the oral cavity for sampling of *A. actinomycetemcomitans* are periodontal pockets, the mucosa, and saliva. Sampling techniques include use of sterile paper points to be inserted in periodontal pockets, cotton swab for the mucosa, and chewing on a piece of paraffin for the collection of stimulated saliva. For transport of paper points, the VMGAIII-medium is recommended [49]; samples collected with cotton swap can be transported in a salt buffer or in TE-buffer [50]. Saliva can be transported in tubes without additives. For short-time transportation, saliva can be transported in tubes without additives. Otherwise, it can be frozen or stored at room temperature in a Saliva DNA Preservation Buffer. Proteomic analysis of gingival crevicular fluid and saliva is an expanding diagnostic field that may require improvements in standardised collection techniques and devices [51,52].

The selective medium TSBV (tryptic soy-serum-bacitracin-vancomycin) agar [53] is commonly used for culture. If *Enterobacterales* are present in significant amounts in the samples, a modified version of TSBV is recommended [54]. Detection of *A. actinomycetemcomitans* in clinical samples renders limited information on prediction, progression, and treatment planning of periodontal disease. For these purposes, the proportion of the bacterium at diseased sites is more relevant. This is in line with the ecological plaque hypothesis [55]. The detection level of *A. actinomycetemcomitans* is around 100 viable bacteria (colony-forming units) per mL. *Fusobacterium nucleatum* and other strict anaerobes will grow on TSBV in the absence of oxygen. The total concentration of viable bacteria is estimated by parallel cultivation on 5% blood agar plates, and the proportion of *A. actinomycetemcomitans* in the sample can be calculated.

*A. actinomycetemcomitans* is suspected when rough-textured, tenacious colonies appear on selective agar after one or two days (Figure 2). The species is distinguished from closely related bacteria by a positive catalase reaction and negative β-galactosidase reaction. Salient biochemical characters of *A. actinomycetemcomitans* have been published [56]. In addition, the bacterium is readily identified by MALDI-TOF mass spectrometry [57]; however, the current version of the Bruker database (v3.1) only includes mass spectra from a limited number of strains, and modest log-scores are not unusual when clinical strains are examined.

Leukotoxicity, i.e., the capacity of the bacterium to kill or inactivate immune cells, is properly determined in biological assays involving human cell lines [58], but a semi-quantitative method based on hemolysis on blood agar plates has been reported [42,59]. Quantification of the leukotoxin by enzyme-linked immunosorbent assay (ELISA) is also used; most studies have assessed the leukotoxin released from the surface of the bacterium, either during growth in broth [60], or by treatment of bacteria cultured in media that inhibit leukotoxin release with a hypertonic salt solution [42]. Leukotoxicity may also be estimated by determination of the total amount of leukotoxin produced by the strain. Bacterial suspensions are solubilized by SDS, and the leukotoxin is subsequently quantitated by Western blot–based methodology [60]. It is anticipated that the amount of leukotoxin released from the bacterial cell surface reflects the total amount of leukotoxin produced, but this relationship remains to be corroborated.

Polymerase chain reaction (PCR) is frequently used for identification and characterisation of *A. actinomycetemcomitans* in clinical samples. The leukotoxin promoter was an early focal point [59]. PCR amplification of the *ltx* promoter region and visualization on gel can discriminate the JP2 genotype from other strains of the species [61], but preferential amplification of smaller products characterised by a 530-bp deletion will overestimate the prevalence of the JP2 genotype. Recent improvements in PCR offer more precise quantification of periodontal pathogens in a complex plaque biofilm [62]. By real-time or quantitative PCR (qPCR), the instrument reports the cycling threshold (CT)-value, which can estimate the concentration of the target in the sample. qPCR has been used to separately quantitate JP2 and non-JP2 genotypes [63]. To approximate the total number of bacteria by qPCR, the 16S rRNA gene is generally targeted. The method can only provide a rough estimate, as primers and probes may preferentially bind to certain bacterial phyla, and because the number of copies of the gene varies substantially between different bacterial species [64].

Serotypes a through f can be identified by PCR as described [65,66]. A method for detecting St(g) has not yet been described.

### 3.2. Aggregative Properties and the Leukotoxin Gene Operon

*A. actinomycetemcomitans* expresses three potential toxins, fimbriae and a number of adhesins, plus a number of other gene products that may have significance for microbial interplay, persistence, transformation to planktonic state, and pathogenicity (Table 2).

The distinct growth in broth as small granules adhering to the walls of the test tube was included in the initial description of [*Bacterium actinomycetem comitans*] [1]. Fresh isolates of *A. actinomycetemcomitans* invariably form colonies that are rough-textured with an opaque, star-shaped internal structure (Figure 2B). Subculture in broth yields clumps of autoaggregated cells that attach tightly to the glass, leaving a clear broth. *A. actinomycetemcomitans* possesses fimbriae, and these appendages can be irreversibly lost after prolonged subculture in the laboratory [77]. Antibodies to synthetic fimbrial peptide significantly reduce the binding of *A. actinomycetemcomitans* to saliva-coated hydroxyapatite beads, buccal epithelial cells and a fibroblast cell line, indicating a decisive role of these structures for adherence to multiple surfaces [78]. Moreover, autoaggregation (spontaneous formation of aggregates with rapid settling in un-agitated suspensions) was completely lost by a smooth-colony, isogenic variant [79]. Fimbriae are assembled as bundles of 5- to 7-nm-diameter pili composed of a 6.5 kDa protein designated Flp (fimbrial low-molecular-weight protein) [80,81]. The RcpA/B (rough colony proteins) were the first outer membrane proteins identified that were associated with rough colony variants [82], and they are encoded by a 14-gene locus designated the *tad* locus. The Tad (tight adherence) macromolecular transport system is a subtype of type II secretion. The *tad* locus is composed of nine *tad*, three *rcp* and two *flp* genes [67]. Mutation analysis of the naturally competent strain D7S revealed *flp-1*, *rcpA*, *rcpB*, *tadB*, *tadD*, *tadE* and *tadF* to be indispensable for expression of fimbriae, while mutants of five other genes expressed reduced levels of fimbriae, or fimbriae that had different gross appearance [83,84]. In a rat model, the *tad* locus was critical for colonizing the oral cavity and for pathogenesis, measured as maxillae bone loss and *A. actinomycetemcomitans*-specific IgG levels [85].

Many pathogenic bacteria can undergo phase variation, but smooth-to-rough conversion has not been substantiated for *A. actinomycetemcomitans*. Rather, the rough-to-smooth conversion is typically caused by mutations in the *flp* promoter region, and replacement with wild-type promoter can restore the rough phenotype [86]. However, one study indicated that smooth strains could re-express the fimbriae in low humidity environments [87].

In addition to expression of fimbriae decisive for autoaggregation and adherence to a wide range of solid surfaces (biofilm formation), *A. actinomycetemcomitans* encodes a spectrum of autotransporter adhesins, proteins that promote their own transport from the periplasm to the exterior surface, where they may be decisive for adhesion to specific human cellular epitopes. A homologue with similarity to the monomeric *H. influenzae* autotransporter, Hap, was designated Aae. Inactivation of *aae* in two naturally transformable strains caused a 70% reduction in adhesion to cultured epithelial cells [68]. Aae exhibits specificity for buccal epithelial cells from humans and Old World primates, and does not bind to human pharyngeal or cervical epithelial cells [88]. Two trimeric autotransporters with homology to the YadA adhesin/invasin family were identified. Omp100 has also been designated Api (*Aggregatibacter* putative invasin). *Escherichia coli* expressing ApiA bound to various types of human collagen plus fibronectin. Adhesion to human cells was specific to buccal epithelial cells from humans and Old-World primates, although the specificity was not as prominent as observed for AaE [70,89]. Screening of a large number of insertion transposon mutants identified the extracellular matrix adhesin A encoded by *emaA*, which is involved in collagen adhesion [90]. Collagen prevail in the supporting tissue of cardiac valves, and EmaA (extracellular matrix adhesin) may play a role in the pathogenesis of infective endocarditis [91].

Iron is an essential transition metal for nearly all forms of life. The host limits the availability of iron through a process termed nutritional immunity [92]. Haemolysis can be an initial step for release of iron from heme by Gram-negative bacteria. The RTX (repeats in toxin) family is an important group of toxins, whose name refers to glycine- and apartate-rich, calcium-binding repeats in the carboxy terminus of the toxin proteins [93]. RTX toxins are produced by many Gram-negative bacteria including members of family *Pasteurellaceae* – it has, indeed, been proposed that these toxins may originate in *Pasteurellaceae* [94].

In 1977, it was shown that polymorphonuclear leukocytes exposed to gingival bacterial plaque in vitro released lysosomal constituents [95], and the leukotoxin (Ltx) of *A. actinomycetemcomitans* was extracted and partially characterised in 1979 [6]. Ltx is a RTX cytolysin. By 1989, the gene was cloned and analysed [96,97], and the 530-bp deletion in the *ltx* promoter associated with enhanced expression of Ltx characterising the JP2 genotype was subsequently described [10]. The difference between minimally toxic and highly toxic strains were convincingly illustrated in clinical studies from Northern Africa [12]. The significance of the 530-bp deletion may reside in a potential transcriptional terminator spanning 100 bp [60]. The leukotoxin of *A. actinomycetemcomitans* is highly specific for human and primate white blood cells and is capable of neutralising local mucosal immune responses. However, purified leukotoxin can lyse sheep and human erythrocytes in vitro, and beta-haemolysis can be demonstrated on certain media [98]. 

In addition to the JP2 genotype characterised by the 530-bp promoter deletion, two other leukotoxin promoter variants have been reported. One genotype is characterised by a slightly enlarged (640-bp) deletion [99], while the other promoter variant carries an 886-bp insertion sequence [100]. Both these variants produce levels of leukotoxin similar to the JP2 genotype of *A. actinomycetemcomitans*.

### 3.3. Geographic Dissemination of Specific Genotypes 

The JP2 clone of *A. actinomycetemcomitans* is suggested to have arisen 2400 years ago in the northern Mediterranean part of Africa [37]. The bacterial clone is endemically present in Moroccan and Ghanaian populations [12,101] and almost exclusively detected among individuals of African origin [37,102]. However, among 17 JP2 clone carriers, living in Sweden and identified during 2000–2014, ten were of Scandinavian heritage [31]. Among six of the identified JP2 clone carriers, three were of Swedish origin. Detection of the JP2 clone of *A. actinomycetemcomitans* has not been reported in Asian populations [30,100,103,104]. The occurrence of the JP2 clone of *A. actinomycetemcomitans* in Caucasians may be caused by horizontal transmission, and may weaken the theory of racial tropism of the clone [59]. More data and research are needed to explain the dissemination of the leukotoxic JP2 clone of *A. actinomycetemcomitans.*

Other genotypes characterised by an increased leukotoxic potential comprise a 640-bp deletion cultured from a host of Ethiopian origin [99], an 886-bp insertion sequence from a host of Japanese origin [100], and two strains of serotype c, originating from Thailand with a JP2-like deletion in the promoter region of *ltx*, and with virulence of similar magnitude to the JP2 genotype strains [105]. All these genotypes were collected from individuals with severe periodontitis.

## 4. Prevalence and Clinical Significance

Cultivable *A. actinomycetemcomitans* is present in at least 10% of periodontally healthy children with primary dentition [106]. An influential publication found carrier rates of 20% for normal juveniles, 36% for normal adults, 50% for adult periodontitis patients, and 90% for young periodontitis patients [107]. Early studies failed to culture the species from edentulous infants [108,109], but molecular studies using PCR on unstimulated saliva samples have challenged this association: 37 of 59 completely edentulous infants were positive for *A. actinomycetemcomitans*, reaching 100% at 12 months of age [110]. Vertical transmission is common. Two studies reported detection rates by culture of 30–60% in children of adult periodontitis patient, and the genotypes of the strains were always identical [111,112]. A smaller study from Brazil of women with severe chronic periodontitis did not corroborate this finding, as the two culture-positive children carried genotypes that were different from those of their mothers [113]. Horizontal transmission of *A. actinomycetemcomitans* can occur, and transmission rates between 14% and 60% between spouses have been estimated [18,114]. However, members of most families with aggressive periodontitis also harbour additional clonal types of *A. actinomycetemcomitans* [115].

Once colonized, *A. actinomycetemcomitans* remains detectable in patients with periodontitis. Irrespective of periodontal treatment, colonisation by the same strain is remarkably stable within subjects for periods of 5 to 12 years, as revealed by restriction fragment length polymorphism [40], serotyping combined with arbitrarily primed PCR [116], or JP2 clone-specific PCR [117]. Genomic stability during persistent oral infections has been demonstrated by genome sequencing of strains cultured from the same individual 10 years later [118].

The natural habitat of *A. actinomycetemcomitans* is the oral cavity, but *A. actinomycetemcomitans* can be isolated from a variety of oral as well as non-oral infectious diseases, including arthritis, bacteraemia, endocarditis, osteomyelitis, skin infections, urinary tract infections and various types of abscesses [119]. Characterisation of 52 non-oral strains showed similarity to oral strains [120], and the portal of entry for systemic infections is usually the oral cavity [121].

### 4.1. Infective Endocarditis

The oral cavity is the only known habitat of *A. actinomycetemcomitans*, but only a few layers of crevicular epithelial cells separate the gingival location from the parenteral space of the host. Entry into the blood stream has not been quantitated, but incidental introductions may occur during tooth brushing, injuries, chewing of granular matters etc., and this may be accelerated by the presence of periodontitis. *A. actinomycetemcomitans* was originally co-isolated with *Actinomyces* from actinomycotic lesions [1], and the association with *Actinomyces* has been confirmed by case reports of infections in a variety of anatomical localizations. Among *Actinomyces* species, co-isolation of *A. actinomycetemcomitans* appears restricted to *Actinomyces israelii* [122,123]. 

Infective endocarditis is an infection of the endocardium, the lining of the interior surfaces of the chambers of the heart. It usually affects the heart valves (Figure 3A), where corrosion and incidental exposure of sub-endothelium tissue during the extensive motion of the valves may serve as a starting point for bacterial adhesion. 

*A. actinomycetemcomitans* is part of the *Haemophilus, Aggregatibacter, Cardiobacterium, Eikenella*, and *Kingella* (HACEK) group of fastidious Gram-negative bacteria that is responsible for 1.4–3% of cases infective endocarditis [124,125]. The group originally included *Haemophilus* species, *Actinobacillus actinomycetemcomitans*, *Cardiobacterium hominis*, *Eikenella corrodens*, and *Kingella kingae* [7]. The HACEK acronym is still valid, but currently denotes non-*influenzae Haemophilus* sp., *Aggregatibacter* sp., *Cardiobacterium* sp., *E. corrodens*, and *Kingella* sp. [126]. *A. actinomycetemcomitans* is the HACEK organism most strongly associated with infective endocarditis [121,125], and bacteraemia with *A. actinomycetemcomitans* necessitates clarification of this putative focus of infection. In a retrospective study of 87 cases of HACEK bacteraemia from New Zealand, the association between HACEK bacteraemia and infective endocarditis varied among bacterial species ranging from 0% (*E. corrodens*) to 100% (*A. actinomycetemcomitans*) [127]. Specific features of infective endocarditis caused by *A. actinomycetemcomitans* have been reviewed [121].

### 4.2. The Complex Interplay with Periodontitis

Periodontitis is an inflammatory disease associated with loss of connective tissue and bone around teeth (Figure 3B). The bacterial tooth biofilm initiates the gingival inflammation, and further progression of the periodontal lesion depends on dysbiotic ecological changes within the gingival sulcus area. Unfavourable lifestyles and hygiene contribute to the development and progression of periodontitis, which has been designated as one of mankind’s most common chronic inflammatory diseases [48]. 

The complexity of the periodontal microbiota and the variety of clinical symptoms delayed the identification of specific microbial aetiological agents. In 1996, *A. actinomycetemcomitans*, *Porphyromonas gingivalis*, and *Tannerella forsythia* were officially designated as aetiological agents of periodontitis [128]. *A. actinomycetemcomitans* was targeted based on prevalence studies in health and disease, serum antibody levels, and presence of virulence determinants (Table 1 and Table 2). More recently, attention has been directed to the complex interplay between other cultivable and other non-cultivable bacteria in the oral microbiota, as well as to the interplay with the host [16,48,129,130]. Indeed, it has been suggested that *A. actinomycetemcomitans* conducts its business by concealing itself from the scrutiny of the host immune system, or even being a community activist that suppresses host responses to allow overgrowth of its collaborators [19].

The earlier classification of aggressive periodontitis was based mainly on the clinical presentation and the rapid loss of periodontal tissue [131]. A new classification scheme has been adopted, in which chronic and aggressive forms of the disease are now merged into a single category, which is characterised by a multi-dimensional staging and grading system [132,133]. Staging assesses severity and extent of disease at presentation, and attempts to include the complexity of disease management. The grading provides an evaluation of the risk of progression, and attempts to predict response to standard periodontal therapy [132].

## 5. Therapy

Treatment of periodontitis aims to stop the progression of the periodontal lesion and to maximise periodontal health [134]. Mechanical debridement of biofilm is considered the most effective therapy, but must be combined with a detailed oral hygiene. If periodontal lesions persist after 3–6 months, a second phase of therapy is planned. A favourable healing potential has been documented for lesions associated with the rapidly progressive, localised periodontitis that affects adolescents [135]. Systemic antibiotics should only be administered as adjunctive therapy in selected cases.

Access surgery with regenerative techniques have been used for periodontitis stages III–IV [132,134]. Notable risk factors are non-compliance, smoking, elevated gingival bleeding index, and inadequate plaque control [136].

Very different amoxicillin resistance rates have been reported, ranging from 0% in Switzerland [137], over 33% in Spain [138] to 84% in the United Kingdom [139]. The mechanisms of resistance were not reported. Production of β-lactamase is the most common cause of β-lactam resistance in Gram-negative bacteria, but these enzymes have not been detected in *A. actinomycetemcomitans*. The fastidious nature of the bacterium is a challenge for antimicrobial susceptibility testing, and methodology as well as interpretative criteria must be addressed when reports of resistance are evaluated. A recent investigation using different methods could not confirm the emergence of resistance to β-lactams in *A. actinomycetemcomitans*; the study included strains that had previously been reported as resistant [140]. Thus, there is currently no convincing evidence for replacement of oral amoxicillin when antimicrobial agents are indicated for treatment of *A. actinomycetemcomitans*-associated periodontitis.

Gram-negative bacteria are generally more susceptible to the cephalosporin-class than the penicillin-class of β-lactams. For infective endocarditis, an intravenous course of at least four weeks with a third-generation cephalosporin, or a combination of ampicillin and an aminoglycoside, is recommended [121]. Recently, a well-designed randomised study reported favourable outcomes for oral antimicrobial follow up regimens given to patients with infective endocarditis deemed clinically stable and without complications [141]. *A. actinomycetemcomitans* could be a candidate microorganism for use of partial oral antimicrobial treatment of infective endocarditis, but the relative rare occurrence of HACEK bacteraemia poses difficulties for additional clinical studies.

## 6. Conclusions

*A. actinomycetemcomitans* is part of the human microbiota. It can be cultured from one-third of healthy adults, while PCR-based methods suggest a more ubiquitous presence. The bacterium’s tenacious, aggregative character is instrumental for the remarkable genotype stability in colonised hosts, and for progression to persistent, distant infections after incidental entry into the parental space. *A. actinomycetemcomitans* are commonly detected if adolescents present with periodontitis. Chronically inflamed gingival crevices may spark the repeated, intermittent entry into the blood stream. Its participation in the disease process of periodontitis is beyond reasonable doubt, but its orchestration of severe periodontitis continues to be fascinating and disputed. Adhesion and leukotoxic features are well-described, but interplay with other members of the oral microbiota is more difficult to elucidate, as is the interchangeable position of eliciting antibody response and “staying under the radar”. The recent division into three subspecies or lineages has not been investigated by clinical studies linking disease and lineage. Disease-specific treatment options are currently widely accepted.

## Figures and Tables

**Figure 1 pathogens-08-00243-f001:**
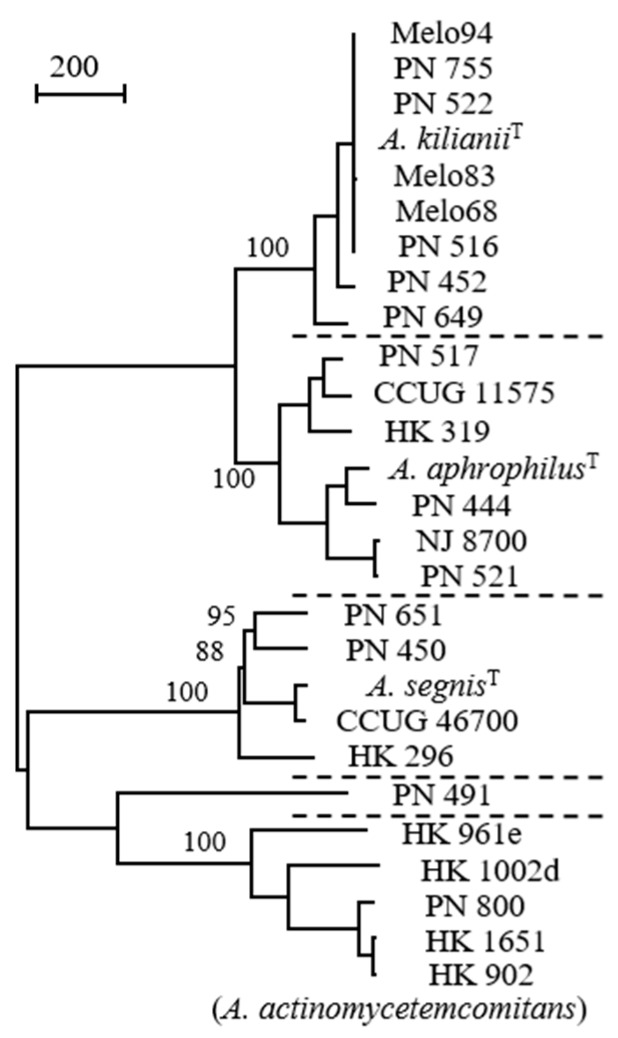
Comparison of *Aggregatibacter* strains by whole genome sequences; distinct species are separated by dotted lines (strain PN_491 is unclustered). A total of 3261 positions with single nucleotide polymorphism (SNP) are included in the dataset. Values at nodes are percentages of bootstrap replications supporting the node (500 replicates). Bar represents 200 SNPs. Reprinted from *Journal of Clinical Microbiology* [24] with permission.

**Figure 2 pathogens-08-00243-f002:**
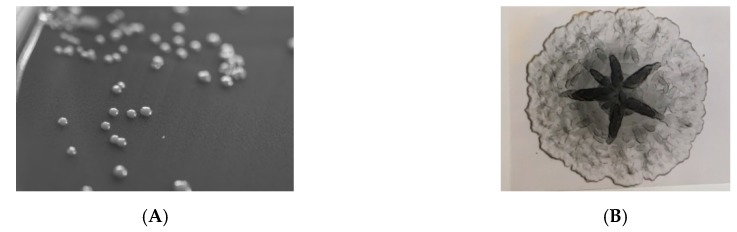
(**A**) Tenacious, rough-textured colonies of *A. actinomycetemcomitans* strain HK1651 on chocolate agar. Diameter of colonies did not reach 2 mm after 3 days incubation in 5% CO_2_. (**B**) Clinical isolate incubated on TSBV (tryptic soy-serum-bacitracin-vancomycin) agar for 4 days in 5% CO_2_. Expression of the distinctive “star-shaped” colony is facilitated by growth on TSBV agar. Pictures by courtesy of Jan Berg Gertsen and Rolf Claesson.

**Figure 3 pathogens-08-00243-f003:**
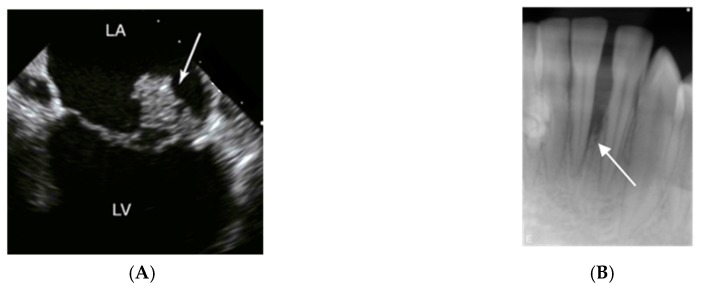
Imaging signs of infections and inflammation that may be associated with *A. actinomycetemcomitans*. (**A**) Transesophageal echocardiography of a heart with mitral valve infective endocarditis. The arrow marks a large vegetation on the posterior leaflet between left atrium (LA) and left ventricle (LV); usually, vegetations caused by *A. actinomycetemcomitans* are of smaller size. (**B**) 14-year old girl of African ethnicity. The radiograph shows an extensive and apparently rapid loss of the periodontal support of the lower incisor 31. Pictures by courtesy of close clinical collaborators of the authors.

**Table 1 pathogens-08-00243-t001:** Seminal events in the history of *Aggregatibacter actinomycetemcomitans*.

Year	Event	Reference
1912	Klinger describes [*Bacterium actinomycetem comitans*]	[1]
1929	Topley and Wilson relocate the species to genus *Actinobacillus*	[2]
1962	King and Tatum describe the close phenotypic similarity of [*Actinobacillus actinomycetemcomitans*] with [*Haemophilus aphrophilus*]	[3]
1976	[*Actinobacillus actinomycetemcomitans*] is associated with periodontitis in adolescents	[4,5]
1979	Extraction and partial characterisation of Ltx, a leukotoxin capable of specific lyse of human polymorphonuclear leukocytes	[6]
1982	The *Haemophilus*, *Aggregatibacter*, *Cardiobacterium*, *Eikenella*, and *Kingella* (HACEK) group of fastidious, Gram-negative bacteria causing infective endocarditis, is conceived	[7]
1982	Serum antibody levels link [*Actinobacillus actinomycetemcomitans*] with localized juvenile periodontitis	[8]
1983	Three distinct surface antigens are identified and a particularly high periodontopathogenic potential of serotype b is indicated	[9]
1994	The 530-bp deletion in the *ltx* promoter region is associated with enhanced expression of Ltx and becomes a marker for the so-called JP2 genotype of [*Actinobacillus actinomycetemcomitans*]	[10]
2006	A new bacterial genus, *Aggregatibacter* is created with *Aggregatibacter actinomycetemcomitans* being the type species	[11]
2008	Clinical follow-up studies unequivocally demonstrate that carriage of the JP2 clone of *Aggregatibacter actinomycetemcomitans* is linked with aggressive periodontitis	[12]

**Table 2 pathogens-08-00243-t002:** Genomic characteristics and putative virulence determinants of *A. actinomycetemcomitans.*

Genomic Characteristics and Putative Virulence Determinants of *A. actinomycetemcomitans*	References
Widespread colonization island or the *tad* locus	[67]
Autotransporter adhesins Aae, EmaA and Omp100/ApiA	[68,69,70]
A leukotoxin of the repeats in toxin (RTX) family with specificity for leukocytes	[71]
Growth-inhibitory factor cytolethal distending toxin	[72,73]
CagE, capable of inducing apoptosis	[74]
Dispersin B, a biofilm-releasing beta-hexosaminidase	[75]
Vesicle-independent extracellular release of proinflammatory lipoprotein	[76]

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
