# Peer review of "Aggregatibacter Actinomycetemcomitans: Clinical Significance of a Pathobiont Subjected to Ample Changes in Classification and Nomenclature"

_pathogens, 2019, doi:10.3390/pathogens8040243_

Round 1

Reviewer 1 Report

The review is written in a comprehensible manner. Minor language corrections needed.

Author Response

Reviewer 1

The review is written in a comprehensible manner. Minor language corrections needed

Reply: Reformulations have been introduced throughout.

Reviewer 2 Report

I have peer reviewed the manuscript titled “Aggregatibacter actinomycetemcomitans: Clinical Significance of a Pathobiont Subjected to Ample Changes in Classification and Nomenclature” submitted to “Pathogens”.

The authors have reviewed the clinical significance, classification, and characterisation of A. actinomycetemcomitans, mainly in context of host specificity for humans. The manuscript is comprehensive, very detailed and fit well with in the scope of this journal. The manuscript needs some major improvements; there are a few suggestions that authors may consider to improve it further:

The abstract is unstructured; however; brief and to the point.

There are a number of grammatical errors and unclear statements; for instance;

Line 35-37: are not clear grammatically; please consider rephrasing it.

Line 35-43: what is the source of this information? Similarly, a number of other statements (including one at Line 73-77) have not been supported by the appropriate citation.

Please define JP2?

Line 82: “number of influential publications or pertinent reviews”; could authors please cite the referred studies?

Introduction: is very detailed; however missing details of periodontitis in context: authors should include more details about the association of periodontitis and A. actinomycetemcomitans (Pathogenesis).

Figure 2 captain is not self-explanatory: please add clear description of figure 2A and 2B?; is this figure reproduced?

For lines 210-212: “Samples collected with cotton swap can be transported in a salt buffer [50] or in TE-buffer. Saliva can be transported in tubes without additives” what about saliva collection devices and their benefits? Including benefits of such devices can be useful for reader; please see the following article that has full details in context;

Human saliva collection devices for proteomics: An update." International journal of molecular sciences 17.6 (2016): 846.

Additionally, there is no details about gingival crevicular fluids (GCF); following article is very relevant and can be useful

Human gingival crevicular fluids (GCF) proteomics: an overview. Dentistry journal. 2017 Mar;5(1):12.

Line 254: please check for grammar; briefly: what were the finding of these studies used PCR.

Please define all abbreviations including HACEK

Author Response

I have peer reviewed the manuscript titled “Aggregatibacter actinomycetemcomitans: Clinical Significance of a Pathobiont Subjected to Ample Changes in Classification and Nomenclature” submitted to “Pathogens”.

The authors have reviewed the clinical significance, classification, and characterisation of A. actinomycetemcomitans, mainly in context of host specificity for humans. The manuscript is comprehensive, very detailed and fit well with in the scope of this journal. The manuscript needs some major improvements; there are a few suggestions that authors may consider to improve it further:

The abstract is unstructured; however; brief and to the point.

Reply: We have continued the unstructered form, but extended the abstract with a sentence referring to the context of the complete oral microbiota.

There are a number of grammatical errors and unclear statements; for instance;

Line 35-37: are not clear grammatically; please consider rephrasing it.

Reply: The sentence has been rewritten [L. 34-36].

Line 35-43: what is the source of this information? Similarly, a number of other statements (including one at Line 73-77) have not been supported by the appropriate citation.

Reply: Lines [30-38] is a short introduction to Table 1, where seminal references are given. The importance of “Adhesion, persistence, and inactivation of immune cells” was possibly overstated in order to justify or explain the focus on these mechanisms for clinical significance. The sentence has been softened [.. immune cells are probably essential for..] [L. 71].

Please define JP2?

Reply: The JP2 clone is formally introduced in Table 1, and the abbreviation is defined in the List of abbreviations in the end of the MS.

Line 82: “number of influential publications or pertinent reviews”; could authors please cite the referred studies?

Reply: The "publications and reviews  .. of other important biochemical mechanisms" briefly referred to in the last sentence of the introduction [L. 73-5] are cited in appropriate paragraphs and in Table 2. This has now been clarified [.. mechanisms of this bacterial species are listed in the relevant sections.] [L. 75].

Introduction: is very detailed; however missing details of periodontitis in context: authors should include more details about the association of periodontitis and A. actinomycetemcomitans (Pathogenesis).

Reply: We accept this prudent advice; however, the socalled ENAaR group has submitted two reviews to the present issue of Pathogens, and the missing context is largely covered by the other one (Virulence and Pathogenicity Properties of Aggregatibacter actinomycetemcomitans.  Pathogens 2019, 8(4), 222)

Figure 2 captain is not self-explanatory: please add clear description of figure 2A and 2B?; is this figure reproduced?

Reply: Figure 2A and B are original pictures, and this has now been stated [L. 186-7].

For lines 210-212: “Samples collected with cotton swap can be transported in a salt buffer [50] or in

TE-buffer. Saliva can be transported in tubes without additives” what about saliva collection devices and their benefits? Including benefits of such devices can be useful for reader; please see the following article that has full details in context; Human saliva collection devices for proteomics: An update." International journal of molecular sciences 17.6 (2016): 846. Additionally, there is no details about gingival crevicular fluids (GCF); following article is very relevant and can be useful: Human gingival crevicular fluids (GCF) proteomics: an overview. Dentistry journal. 2017 Mar;5(1):12.

Reply: The focus of the review is on bacteria, but the expanding diagnostic field of fluid proteomic analysis is now included citing the two suggested references [L. 196-8].

Line 254: please check for grammar; briefly: what were the finding of these studies used PCR.

Reply: The paragraph describing the use of PCR has been re-written [L. 227-37].

Please define all abbreviations including HACEK

Reply: Additional definitions has now been included in the text, and all definitions are listed in the section before the references

Reviewer 3 Report

This is an outstanding paper describing both the history and biology of Aggregatibacter actinomycetemcomtians by a group of authors well qualified to perform such an extensive review.  The paper is well written, well organized, with outstanding documentation and clear explanation of the state of affairs for this important oral pathobiont.  This review will provide unlimited resources for individuals who are working or who plan to work in this field.  Congrats to the authors for presenting such a comprehensive and well written manusript.

Author Response

This is an outstanding paper describing both the history and biology of Aggregatibacter actinomycetemcomtians by a group of authors well qualified to perform such an extensive review.  The paper is well written, well organized, with outstanding documentation and clear explanation of the state of affairs for this important oral pathobiont.  This review will provide unlimited resources for individuals who are working or who plan to work in this field.  Congrats to the authors for presenting such a comprehensive and well written manusript

Reply: Thank you.

Reviewer 4 Report

This is an excellent written review on Aggregatibacter actinomycetemcomitans. It covers all aspects starting from taxonomy, classification, virulence factors via clinical significance in dentistry and other disciplines of medicine.

I have only a very minor comment. The authors describe within the text that A. actinomycetemcomitans might be present also in periodontally healthy subjects. This should be clarified also in conclusion and abstract.

Author Response

This is an outstanding paper describing both the history and biology of Aggregatibacter actinomycetemcomtians by a group of authors well qualified to perform such an extensive review.  The paper is well written, well organized, with outstanding documentation and clear explanation of the state of affairs for this important oral pathobiont.  This review will provide unlimited resources for individuals who are working or who plan to work in this field.  Congrats to the authors for presenting such a comprehensive and well written manuscript

Reply: Thank you.

Round 2

Reviewer 2 Report

Many thanks for revising the manuscript;